# Orthostatic Symptoms and Reductions in Cerebral Blood Flow in Long-Haul COVID-19 Patients: Similarities with Myalgic Encephalomyelitis/Chronic Fatigue Syndrome

**DOI:** 10.3390/medicina58010028

**Published:** 2021-12-24

**Authors:** C. (Linda) M. C. van Campen, Peter C. Rowe, Frans C. Visser

**Affiliations:** 1Stichting CardioZorg, Planetenweg 5, 2132 HN Hoofddorp, The Netherlands; fransvisser@stichtingcardiozorg.nl; 2Department of Pediatrics, Johns Hopkins University School of Medicine, Baltimore, MD 21287, USA; prowe@jhmi.edu

**Keywords:** long-haul COVID-19, chronic fatigue syndrome (CFS), myalgic encephalomyelitis (ME), cerebral blood flow (CBF), orthostatic intolerance, tilt testing, postural orthostatic tachycardia syndrome (POTS)

## Abstract

*Background and Objectives*: Symptoms and hemodynamic findings during orthostatic stress have been reported in both long-haul COVID-19 and myalgic encephalomyelitis/chronic fatigue syndrome (ME/CFS), but little work has directly compared patients from these two groups. To investigate the overlap in these clinical phenotypes, we compared orthostatic symptoms in daily life and during head-up tilt, heart rate and blood pressure responses to tilt, and reductions in cerebral blood flow in response to orthostatic stress in long-haul COVID-19 patients, ME/CFS controls, and healthy controls. *Materials and Methods*: We compared 10 consecutive long-haul COVID-19 cases with 20 age- and gender-matched ME/CFS controls with postural tachycardia syndrome (POTS) during head-up tilt, 20 age- and gender-matched ME/CFS controls with a normal heart rate and blood pressure response to head-up tilt, and 10 age- and gender-matched healthy controls. Identical symptom questionnaires and tilt test procedures were used for all groups, including measurement of cerebral blood flow and cardiac index during the orthostatic stress. *Results*: There were no significant differences in ME/CFS symptom prevalence between the long-haul COVID-19 patients and the ME/CFS patients. All long-haul COVID-19 patients developed POTS during tilt. Cerebral blood flow and cardiac index were more significantly reduced in the three patient groups compared with the healthy controls. Cardiac index reduction was not different between the three patient groups. The cerebral blood flow reduction was larger in the long-haul COVID-19 patients compared with the ME/CFS patients with a normal heart rate and blood pressure response. *Conclusions*: The symptoms of long-haul COVID-19 are similar to those of ME/CFS patients, as is the response to tilt testing. Cerebral blood flow and cardiac index reductions during tilt were more severely impaired than in many patients with ME/CFS. The finding of early-onset orthostatic intolerance symptoms, and the high pre-illness physical activity level of the long-haul COVID-19 patients, makes it unlikely that POTS in this group is due to deconditioning. These data suggest that similar to SARS-CoV-1, SARS-CoV-2 infection acts as a trigger for the development of ME/CFS.

## 1. Introduction

From the early stages of the SARS-CoV-2 virus pandemic, it has become clear that affected subjects can experience long-lasting, exhausting fatigue accompanied by post-exertional malaise (PEM), cognitive dysfunction, and an early-onset symptoms of orthostatic intolerance [1,2]. In a large, international survey of adults with persistent symptoms following suspected or confirmed COVID-19 illness, the three most commonly reported symptoms were fatigue, PEM, and cognitive dysfunction [2]. Several studies have identified orthostatic intolerance—defined as a group of clinical conditions in which symptoms worsen upon assuming and maintaining upright posture and are ameliorated by recumbency [3,4]—after COVID-19, with particular emphasis on the early-onset of postural orthostatic tachycardia syndrome (POTS) [2,5,6,7,8,9]. 

These chronic symptoms following COVID-19 illness are also core symptoms of myalgic encephalomyelitis/chronic fatigue syndrome (ME/CFS) [4,10,11]. Fatigue is the cardinal symptom in CFS, post-exertional malaise in ME. Cognitive dysfunction is a highly prevalent and classifying symptom in both the CFS and ME criteria [12,13]. Orthostatic intolerance is also highly prevalent, affecting up to 90% of adults and >95% of pediatric patients in the ME/CFS population [14,15]. Petracek et al. reported that their patients satisfied criteria for ME/CFS six months after SARS-CoV-2 infection, suggesting that the virus may trigger ME/CFS, similar to SARS-CoV-1 and other viral infections [16,17,18,19,20].

To further explore whether the similarity in chronic symptoms of patients after SARS-CoV-2 infection (long-haul COVID-19) and ME/CFS extends to objective hemodynamic abnormalities, we measured orthostatic symptoms in daily life and during head-up tilt, heart rate and blood pressure responses to tilt, and reductions in cerebral blood flow during orthostatic stress testing, comparing 10 consecutive long-haul COVID-19 patients to 20 ME/CFS patients without POTS, and 20 ME/CFS patients with POTS. 

## 2. Materials and Methods

### 2.1. Participants

The case-control study was conducted in the outpatient clinic of the Stichting CardioZorg, a cardiology clinic in the Netherlands that specializes in diagnosing and treating adults with ME/CFS. Cases were eligible if the start of their illness was associated with confirmed SARS-CoV-2 (COVID-19) infection, or if their acute symptoms were consistent with SARS-CoV-2 in early 2020 (from February to June 2020), during a period where COVID-19 testing was not recommended, or not available. For probable cases, the clinical diagnosis of SARS-CoV-2 virus infection had to be made by pulmonologists, internists, general practitioners, and doctors of the GGD (Municipal Health Service). 

The controls were identified from the clinic database of ME/CFS patients who visited our clinic between November 2015 and July 2021, in whom a tilt test was performed for quantification of orthostatic intolerance (OI). ME/CFS controls were matched to the long-COVID-19 cases first by gender, then by age (+/− 1 years), selecting the closest matching patient to the case. Because our previous research [21] showed that patients with hypermobility have a larger cerebral blood flow reduction during head-up tilt, and with 3/10 of long-haul COVID-19 patients being diagnosed with hypermobility, we also ensured that 30 percent or the ME/CFS patients also met criteria for joint hypermobility. It was not possible to match this variable by gender, given the small number of hypermobile males in our ME/CFS database. Patients were considered hypermobile if the diagnosis of joint hypermobility, or hypermobile Ehlers-Danlos Syndrome (hEDS) had been made by a geneticist, rheumatologist, or specialized rehabilitation physician. In all other patients seen during the study period, in whom a formal diagnosis of hypermobility had not been established, we asked whether they were highly flexible or were hypermobile. In the event of a positive answer, the Beighton score was obtained [22]. For this study, we chose a conservative, elevated Beighton score of 6 or higher as the threshold for confirming the diagnosis of hypermobility [22,23]. The diagnosis of ME/CFS was made according to the ME/CFS criteria of the International Consensus Criteria (ICC) [10,11], and we excluded those with any other illnesses that could explain the symptomatology. Moreover, we scored ME/CFS symptoms based on the Institute of Medicine (IOM) criteria [4]. We noted if study participants were using medications that could alter heart rate (HR) or blood pressure (BP), and these drugs were discontinued before performing the tilt test. The ME/CFS controls became ill at least 2 years before the start of the SARS-CoV-2 pandemic. In light of the reported occurrence of POTS following COVID-19, we selected 20 ME/CFS controls with POTS and 20 without POTS. We also selected a third control group of 20 gender- and age-matched healthy adults without ME/CFS. 

The study was carried out in accordance with the Declaration of Helsinki. All ME/CFS participants and healthy controls gave informed, written consent. The study was approved by the medical ethics committee of the Slotervaart Hospital, Amsterdam, the Netherlands (P1450).

### 2.2. ME/CFS Criteria by International Consensus Criteria, Fukuda and Institute of Medicine 

In order to study whether daily symptoms of long-haul COVID-19 and ME/CFS were comparable, we used a questionnaire to systematically ascertain whether patients satisfied the criteria for CFS described by Fukuda et al. [10], the ME criteria described in the International Consensus Criteria by Carruthers et al. [11], and the ME/CFS criteria derived from the report of the Institute of Medicine [4]. Based on the ME criteria, patients were categorized as having typical ME or atypical ME; based on the Fukuda criteria patients were categorized as having CFS or chronic fatigue; based on the IOM criteria patients were categorized as having ME/CFS present or absent. 

Using questionnaires we have employed in earlier studies for the evaluation of ME/CFS patients, we ascertained the presence of the following symptoms as outlined in the ME/CFS criteria: symptoms present more than 6 months; fatigue/exhaustion; physical and or mental exercise intolerance; symptom increase after physical and/or mental exercise (called PEM); prolonged recovery from PEM; memory problems; headaches; muscle pain; joint pain; unrefreshing sleep; sensory hypersensitivity (light, sound, smells, vibrations, touch); neuromotor impairment (muscle weakness, disturbed coordination, fasciculations, ataxia); flu-like symptoms; sore throat; tender lymph nodes; increased sensitivity to viral infections; gastrointestinal symptoms; genito-urinary symptoms; hypersensitivity to food and chemicals; orthostatic intolerance; respiratory symptoms; thermal instability including sweating, and extreme temperature intolerance. For comparison of symptoms of the long-haul COVID-19 patients, the ME/CFS patients with POTS and with a normal HR and BP response were combined. Moreover, we asked about the triggers that led to the ME/CFS symptoms. 

### 2.3. Tilt Test with Extracranial Doppler Cerebral Blood Flow Measurements

Measurements were performed as described previously [15,24]. Briefly, all participants were positioned for 20 min supine before being having their head tilted up to 70 degrees for a maximum of 30 min (mainly in healthy controls). The process of tilting upright took approximately 30 s. HR, systolic blood pressure, and diastolic blood pressure were continuously recorded by finger plethysmography. Heart rate, systolic blood pressure and diastolic blood pressure were extracted from the device and imported into an Excel spreadsheet. The test ended after 30 min or earlier, in case of orthostatic hypotension or pre-syncope, or at the request of patients if they were unable to tolerate the upright position any longer. In healthy controls two upright measurements were available: at mid-tilt and at the end of the tilt period. As the mid-tilt period of healthy controls was similar to the end-tilt period of patients (13 ± 4 min in healthy controls and 13 ± 7 min in patients), the mid-tilt data of the healthy controls was used for comparison with patients. 

Internal carotid artery and vertebral artery Doppler flow velocity frames were acquired by one operator (FCV), using a Vivid-I system (GE Healthcare, Hoevelaken, The Netherlands) equipped with a 6–13 MHz linear transducer. High resolution B mode images, color Doppler images and the Doppler velocity spectrum (pulsed-wave mode) were recorded in one frame. At least two consecutive series of six frames per artery were recorded. Image acquisition for all 4 vessels at each time point (supine and end-tilt) lasted 3 (1) min. Blood flow of the internal carotid and vertebral arteries was calculated offline by an investigator (CMCvC), who was unaware of the patient case or control status. Blood flow in each vessel was calculated from the mean blood flow velocities x the vessel surface area and expressed in mL/min. Vessel diameters were manually traced by CMCvC on B-mode images, from the intima to the opposite intima. Surface area was calculated as follows: the peak systolic and end diastolic diameters were measured, and the mean diameter was calculated as: mean diameter = (peak systolic diameter × 1/3) + (end diastolic diameter × 2/3) [25]. Flow in the individual arteries was calculated in 3–6 cardiac cycles, and data were averaged. Total cerebral blood flow was calculated by adding the flow of the four arteries. End-tidal PCO_2_ (PETCO_2_) was monitored using a Lifesense device (Nonin Medical, Minneapolis USA).

### 2.4. Doppler Measurements for the Determination of the Cardiac Index

Cardiac Index (CI) is the cardiac output corrected for body surface area. Measurements were performed as described previously [26]. Briefly, the time–velocity integral (VTI) of the aorta was measured using a continuous-wave Doppler pencil probe connected to a Vivid I machine (GE, Hoevelaken, NL) with the transducer positioned in the suprasternal notch. A maximal Doppler signal was assumed to be the optimal flow alignment. At least 2 frames of 6 s were obtained. Time–velocity integral frames were obtained in the resting supine position, and at the end of tilt test phase. From an echocardiogram performed earlier, the diameter of the outflow tract was obtained. The aortic time velocity integral was measured by manual tracing of at least 6 cardiac cycles, using the GE EchoPac post-processing software. This was done by one operator (CMCvC). Stroke volumes indices (SVI) were calculated from the time–velocity integral and the outflow tract area, corrected for the aortic valve area [27,28], and divided by the body surface area (Du Bois formula). Stroke volume index of the separate cycles were averaged. The cardiac index was calculated from the heart rate and stroke volume index. We have previously validated this methodology by a direct comparison with CI measurements using transthoracic time–velocity integral images from the apical 4-chamber view [26].

### 2.5. Hemodynamic Classification of Heart and Blood Pressure Changes during Tilt Testing

The changes in heart rate and blood pressure during the tilt were classified according to the consensus statement and guidelines [29,30,31] as follows: (a) normal heart rate and blood pressure response; (b) postural orthostatic tachycardia syndrome (POTS), defined as a sustained increase in HR of 30 bpm or more within 10 min of standing, without an abnormal blood pressure response; and (c), syncope or near-syncope. For comparison purposes, with COVID-19 cases all having POTS, 20 ME/CFS POTS patients were analyzed as a control group, as well as a group of ME/CFS patients with a normal heart rate and blood pressure response (*n* = 20), and a control group of 20 healthy controls.

### 2.6. Orthostatic Symptoms during Tilt

Orthostatic symptoms during tilt testing were ascertained using a written questionnaire that elicited whether 15 orthostatic intolerance symptoms were present, as described previously [15]. Directly after attaining the upright position, a clinician (FCV) read the questionnaire items to the patients and healthy controls, and participants were asked to respond with yes or no. The questionnaire is available in the Appendix A. 

### 2.7. Statistical Analysis

Data were analyzed using Graphpad Prism version 6.05 (Graphpad software, La Jolla, CA, USA). All continuous data were tested for normal distribution using the D’Agostino–Pearson omnibus normality test, and presented as mean with standard deviation (SD) or as median with the interquartile range (IQR), where appropriate. Nominal data were compared using the Chi-square test. Between-group comparison was done by the one-way analysis of variance (ANOVA). Where significant, results were then explored further using the post-hoc Tukey’s multiple comparison test. In both the long-haul COVID-19 patients and in the ME/CFS patients (groups 2 and 3) disease duration was not normally distributed, and between-group comparison was done using the Kruskal–Wallis test. Where significant, results were then explored using a post-hoc Dunn’s multiple comparison test. Criteria symptom clusters were scored as present or absent. The percentage of present symptoms was calculated for each symptom cluster of the long-haul COVID-19 subjects and for ME/CFS subjects, and 95% confidence intervals were calculated. These nominal data were also compared with Chi-square (2 × 2 table). Due to the multiple comparisons, we chose to use a *p* value of < 0.01 to be significant. 

## 3. Results

The first long-haul COVID-19 patient was evaluated in November 2020. Since then, nine more long-haul COVID-19 patients were seen at the clinic with a disease duration between 6 months and 15 months. Those 10 patients were the cases. Respiratory symptoms were present in 8/10 immediately after the onset of acute infectious symptoms; in two patients respiratory symptoms developed at three weeks after the onset of other acute infectious symptoms. None of the patients were admitted to hospital, and thus none were admitted to the intensive care unit. All 10 long-haul COVID-19 cases had a tilt test consistent with POTS. The onset of orthostatic intolerance/POTS symptoms started 2.1 ± 0.7 weeks after the onset of viral symptoms. From the database of 452 ME/CFS patients with normal heart rate and blood pressure responses, we selected 20 controls, matched on age, then gender. From the database of 302 ME/CFS patients with POTS, we selected 20 controls, also age- and gender-matched. From the database of 61 healthy adults, we selected 20 age- and gender-matched controls. Table 1 shows the baseline demographic and clinical characteristics of the long-haul COVID-19 cases and control participants (ME/CFS patients and healthy controls). The matching process was successful. All ME/CFS patients fulfilled the criteria for CFS and the Institute of Medicine criteria. One long-haul COVID-19 patient and two ME/CFS patients with normal heart rate and blood pressure responses had atypical ME, whereas the others had ME. As expected, disease duration differed significantly between the long-haul COVID-19 patient group and the 2 ME/CFS comparison groups. No difference was found in disease duration between the ME/CFS patients with POTS or ME/CFS patients with a normal heart rate and blood pressure response during the tilt test. 

Table 2 displays the self-reported daily symptom cluster scores for the long-haul COVID-19 cases and the ME/CFS controls. Figure 1 provides a graphical representation of the data. No significant differences were found in any of the ME/CFS symptom clusters between long-haul COVID-19 patients and ME/CFS patients. 

The trigger for the development of ME/CFS symptoms was the confirmed or probable SARS-CoV-2 virus in long-haul COVID-19 patients. In the 40 ME/CFS patients, reported triggers were an infection (bacterial, viral, borreliosis) in 23 (58%), trauma in 3 (8%), surgery in 2 (5%), burnout in 2 (5%), and pregnancy in 1 (3%). The onset of symptoms was insidious in 9 (23%).

Table 3 shows the tilt test measurements of heart rate, systolic blood pressure, diastolic blood pressure, PetCO_2_, cerebral blood flow, and cardiac index for each group. Compared with long-haul COVID-19 cases and the two ME/CFS patient groups, healthy controls had a lower end-tilt heart rate, a lower end-tilt diastolic blood pressure, and a lower supine cardiac index. By definition, the HR at end-tilt in ME/CFS POTS patients was higher than in the ME/CFS patients with a normal heart rate and blood pressure response. The difference in PETCO_2_ (end-tilt minus supine) was significantly less in healthy controls than in the three patient groups. This was also observed for the percent reduction in cardiac index at end-tilt. 

Figure 2 shows the percent reduction in cerebral blood flow (end-tilt minus supine/supine x100%) for the long-haul COVID-19 case group and three control groups. All patient groups had a significantly larger reduction in end-tilt cerebral blood flow compared with the healthy controls. ANOVA analysis among the three patient groups showed a significant difference (*p* = 0.005); the post-hoc analysis identified a significant difference between the long-haul COVID-19 group and the ME/CFS group with a normal heart rate and blood pressure response (*p* = 0.0011). Those with long-haul COVID-19 and POTS, and ME/CFS and POTS, did not differ (*p* = 0.49).

Figure 3 shows the mean number of the 15 questions addressed directly after tilting to the upright position where patients answered with yes. A significant difference was found in the number of complaints between healthy controls and the three patient groups. The long-haul COVID-19 cases and the two ME/CFS patient control groups did not differ on a one-way ANOVA analysis. As palpitations are prominent in the daily life of patients with POTS, we compared the palpitation question from the orthostatic intolerance questionnaire in the three patient groups. Six of 10 in the long-haul COVID-19 group, 8/20 in the ME/CFS POTS group, and 5/20 in the ME/CFS normal heart rate and blood pressure response group reported palpitations (Chi square 3523.2; *p* = 0.17). Similarly, dizziness or lightheadedness were also compared between the three patient groups. All long-haul COVID-19 patients and all ME/CFS patients with POTS reported dizziness/lightheadedness directly after the onset of the upright test phase. In 18/20 ME/CFS patients with a normal heart rate and blood pressure response, dizziness/lightheadedness was reported after the onset of the tilt phase. Figure 4 shows an example of cerebral blood flow in the left carotid artery supine and end-tilt standing. The upper panel shows a long-haul COVID-19 subject and the lower panel a healthy control. Figure 5 shows an example of cardiac output supine and end-tilt standing. The upper panel shows a long-haul COVID-19 subject and the lower panel a healthy control.

## 4. Discussion

The main finding of this study is that all long-haul COVID-19 patients met diagnostic criteria for ME/CFS by 6 months from their initial respiratory illness. Analysis of 21 different symptom clusters that are part of the three different ME/CFS definitions showed no differences in symptom prevalence of long-haul COVID-19 patients compared with the ME/CFS patients (Figure 1). Our study extends previously reported findings by confirming that long-haul COVID-19 patients also have objective and significant reductions in cerebral blood flow during tilt, similar to those with long-standing ME/CFS. In ME/CFS patients with an abnormal cerebral blood flow reduction, by direct or indirect measures [15,32,33,34,35,36,37,38,39], and an abnormal cardiac index reduction during a tilt test, have been previously shown by us and others [40,41].

Surveys exploring the prevalence of 203 symptoms, tracking 66 symptoms in long-haul COVID-19 patients over seven months [2], confirmed that the majority of the symptoms are part of the different ME/CFS criteria. The similarity of symptoms between long-haul COVID-19 and ME/CFS symptoms is indicated as a red X in the Appendix A of the ME, CFS, and Institute of Medicine criteria. Of interest, many of the long-haul COVID-19 symptoms that are not part of the ME/CFS criteria were incidentally mentioned by ME/CFS patients during their visit at our outpatient clinic. Despite the strikingly similar symptoms between long-haul COVID-19 and ME/CFS patients, the authors of the survey reported that only 118/3762 (3%) patients had been diagnosed with ME/CFS and 155/3762 (4%) with POTS. The survey is a self-administered questionnaire, which limits the diagnostic accuracy for ME/CFS, and a prospective study is needed in which the diagnosis of ME/CFS is made by expert clinicians. Furthermore, long-haul COVID-19 patients may benefit from the treatment options already published in ME/CFS patients.

It must be remembered that the diagnostic criteria for ME/CFS have changed significantly over time. For the diagnosis of CFS the Fukuda criteria contained a limited number of symptoms. The ICC criteria for ME included a more expansive list of symptoms, whereas the most recent IOM consensus for ME/CFS simplified the number of criteria when there was an absence of strong scientific proof that specific symptoms were sufficiently prevalent in ME/CFS and when those symptoms were inadequate for discriminating ME/CFS from other diseases [4,10,11]. The present extensive research effort in SARS-CoV-2 symptomatology and the similarity with the ME symptomatology may lead to a reappraisal of the multi-symptom aspect of ME/CFS, which was lost in the Institute of Medicine ME/CFS criteria.

Multiple recent reviews on SARS-CoV-2 have shown a range of neurological manifestations, supported by imaging and a brain autopsy series [42,43,44,45,46,47,48]. This suggests that the inflammation of the central nervous system plays a role in the pathogenesis of the neurological manifestations of the disease. Several studies in the past have suggested a similar neuro-inflammatory basis of ME/CFS, as was shown in several imaging studies and reviews [49,50,51,52]. This similarity—together with the cardiac index and cerebral blood flow reduction found in both long-haul COVID-19 cases and ME/CFS controls—supports the view that long-haul COVID-19, with a symptom duration over 6 months, is a form of ME/CFS, where SARS-CoV-2 presents as a trigger for the development of the disease. However, other authors have questioned this trigger–disease relation, stating that there is currently insufficient evidence to establish COVID-19 as an infectious trigger for ME/CFS [53].

With respect to the presentation of complaints, several points need to be emphasized. First, patients with long-haul COVID-19 usually have had a mild acute SARS-CoV-2 infection. In contrast, more severely ill SARS-CoV-2 patients, in need of hospitalization and respiratory assistance, present with more severe respiratory symptoms such as viral pneumonia, and ultimately acute respiratory distress syndrome (ARDS). This suggests that the pathogenesis of the chronic SARS-CoV-2 form has a neuro-inflammatory basis and is less dependent on primary lung injury. Second, orthostatic intolerance symptoms present early in the COVID-19 illness, in our series between 1 and 3 weeks. Other reports also show that POTS is an early symptom of SARS-CoV-2 [5,9,54]. The series reported from the Mayo Clinic shows that POTS is not the only presentation of dysautonomia and orthostatic intolerance complaints in these patients [54]. Having a normal heart rate and blood pressure response to orthostatic stress is prevalent in ME/CFS patients, occurring in 58% of our population [15]. From the available published data on long-haul COVID-19 orthostatic intolerance cases, with the cases of the current study included, the majority of patients presented with POTS: 42 of 71 patients (59%) [6,7,9,54,55,56]. Given the high number of case reports, publication bias undoubtedly plays an important role. More prospective studies on the incidence of orthostatic intolerance abnormalities in long-haul COVID-19 patients are needed.

In the American Heart Association guidelines on the evaluation of patients with syncope, a distinction is made between POTS and postural tachycardia [31], where POTS is associated with orthostatic intolerance symptoms, and postural tachycardia is not. The orthostatic intolerance symptoms are a complex of symptoms considered to result from cerebral hypoperfusion and/or sympathetic overdrive [3,35]. At present, the objective demonstration of the cerebral hypoperfusion is not part of the definition of POTS in the guidelines, although we have demonstrated this cerebral hypoperfusion in the ME/CFS population and others in other POTS populations [57,58,59,60]. Therefore, we also administered to patients and healthy controls 15 orthostatic intolerance questions directly after onset of the tilt phase. This questionnaire has been previously validated [15]. In all three patient groups (long-haul COVID-19 patients, ME/CFS patients with POTS and with normal HR and BP responses) the number of positively answered questions were significantly higher, compared with the healthy controls (see Figure 3). Therefore, we are confident that our long-haul COVID-19 patients had POTS and not postural tachycardia.

In previous studies, orthostatic intolerance and especially POTS, have been considered to be associated with cardiovascular deconditioning [3,61,62,63]. We recently described cerebral blood flow reductions in ME/CFS patients without deconditioning, mild deconditioning, and with moderate/severe deconditioning as measured by the peak VO_2_ assessed during cardiopulmonary exercise testing (the maximum amount of oxygen you can utilize during exercise). This is commonly used to test the aerobic endurance or cardiovascular fitness of subjects [63]. No significant difference was found in the extent of cerebral blood flow reduction between no, mild, and moderate/severe deconditioning groups; moreover, no difference in VO_2_ was found in ME/CFS patients presenting with POTS, delayed orthostatic hypotension, or with a normal heart rate and blood pressure response on tilt testing [64]. In the present study, in the long-haul COVID-19 patients with POTS, the orthostatic intolerance/POTS complaints began early after the start of the viral infection symptoms (1–3 weeks), consistent with other published case reports [5,7,9,56]. All cases described were patients who were very fit before the start of the infection, participating in sports activities for multiple hours per week. In light of the early-onset of orthostatic intolerance/POTS, and the high physical fitness before the start of the SARS-CoV-2 infection, it is also unlikely in this patient group that deconditioning plays an important role in their orthostatic intolerance and POTS.

An important question is whether the long-haul COVID-19 patients will ultimately recover from the infection. A few studies in ME/CFS have addressed this question [65,66,67,68,69,70]. Complete recovery in adults was observed between 4 and 31%. However, many follow-up studies were limited, in that they either relied on a retrospective self-report at a single point in time, or they consisted of longitudinal data that were analyzed in a cross-sectional manner without taking into account the influence of baseline findings. Moreover, many ME/CFS follow-up studies employed medical care samples instead of random community samples of socioeconomically and ethnically diverse populations [66]. For the long-haul COVID-19 cohort in the study by Davis et al. [2], 257 respondents (6.8%) recovered after day 28 of illness. The data suggest that recovery in the long-haul COVID-19 is limited; however, the follow-up duration so far is short, and longer follow-up data are needed.

### Limitations

We acknowledge that referral bias by the general practitioner may have played a role, selectively referring patients with orthostatic symptoms. Furthermore, as the hypothesis of long-haul COVID-19 as a trigger for ME/CFS is still a topic of discussion, only a small population has been studied as of yet. Larger patient groups need to be prospectively studied, as well as an extensive follow-up of one or more years to determine whether the cerebral blood flow abnormalities disappear in time, together with the recovery of complaints and improvements in physical/mental functioning. Conversely, we did not investigate patients who were partly or completely recovered from the SARS-CoV-2 infection. Finally, in 8/10 patients, the SARS-CoV-2 infection was not confirmed by antigen testing because of the limited availability of tests at that time, when testing was only done in patients presenting at the emergency room, or admitted to hospital. In all 10 patients, the diagnosis of a SARS-CoV-2 infection was made by experienced physicians before they were referred to us.

## 5. Conclusions

This study demonstrates that symptoms of long-haul COVID-19 patients are very similar to ME/CFS symptoms. All long-haul COVID-19 patients had orthostatic intolerance symptoms and developed POTS during a tilt test. The heart rate abnormalities were accompanied by a reduction in both cerebral blood flow and cardiac index. Given the findings of comparable symptoms, hemodynamic, and cerebral blood flow abnormalities in long-haul COVID-19 and ME/CFS patients, our data support the notion that SARS-CoV-2 infection acts as a trigger for the development of ME/CFS in long-haul COVID-19 patients. Furthermore, the very early onset of orthostatic intolerance complaints in a previously physically fit population makes it highly unlikely that POTS in these patients is related to deconditioning.

## Figures and Tables

**Figure 1 medicina-58-00028-f001:**
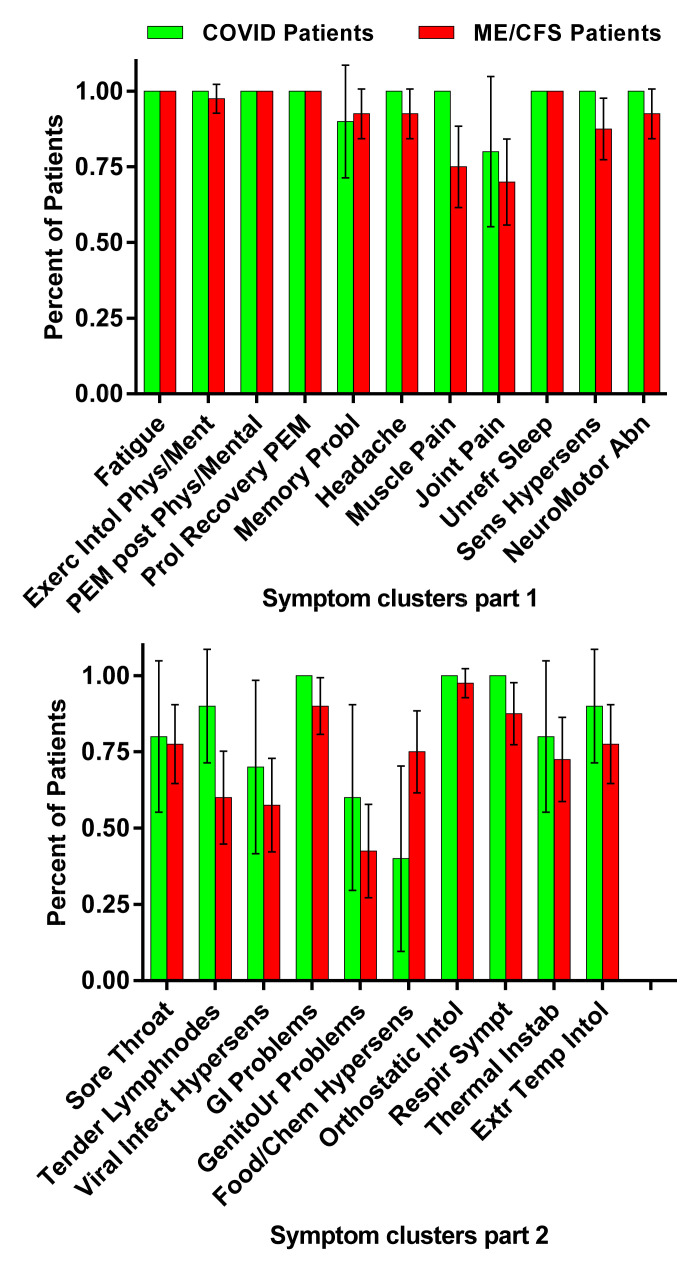
Number of positive symptom clusters derived from ME/CFS/Institute of Medicine criteria in the long-haul COVID-19 patients and all ME/CFS patient. Legend Figure 1 ME/CFS: myalgic encephalomyelitis/chronic fatigue syndrome; Exerc: exercise; Intol: intolerance: Phys: physical; Ment: mental; PEM: post-exertional malaise; Prol: prolonged; Probl: problems; Unrefr: unrefreshing; Sens: sensory; Hypersens: hypersensitivity; Abn: abnormality; Sympt: symptoms: Infect: infection; GI: gastro-intestinal; GenitoUr: genito-urogenital; Respir: respiratory; Instab: instability; Extr: extreme; Temp: temperature.

**Figure 2 medicina-58-00028-f002:**
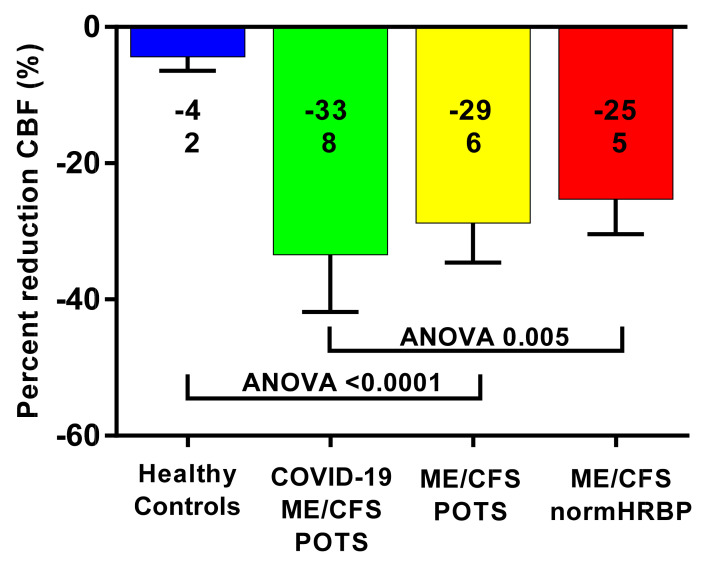
Percent reduction in cerebral blood flow (end-tilt minus supine/supine × 100%) in the long-haul COVID-19 patients, ME/CFS patients with POTS, ME/CFS patients with a normal heart rate and blood pressure response, and healthy controls. Legend Figure 2 CBF: cerebral blood flow; ME/CFS: myalgic encephalomyelitis/chronic fatigue syndrome; BP: blood pressure; HR: heart rate; POTS: postural orthostatic intolerance syndrome.

**Figure 3 medicina-58-00028-f003:**
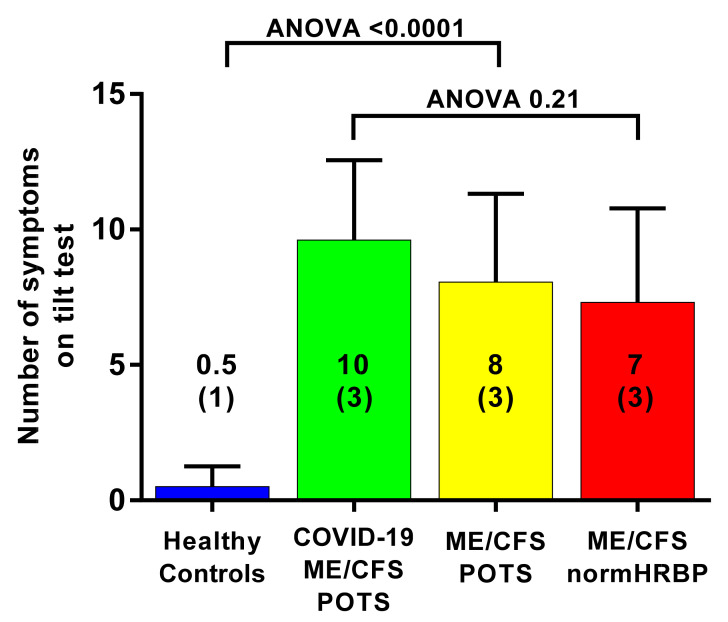
Mean number of positive response to 15 questions, obtained directly after tilting to the upright position, in long-haul COVID-19 patients, ME/CFS patients with POTS, ME/CFS patients with a normal heart rate and blood pressure response, and healthy controls. Legend Figure 3 ME/CFS: myalgic encephalomyelitis/chronic fatigue syndrome.

**Figure 4 medicina-58-00028-f004:**
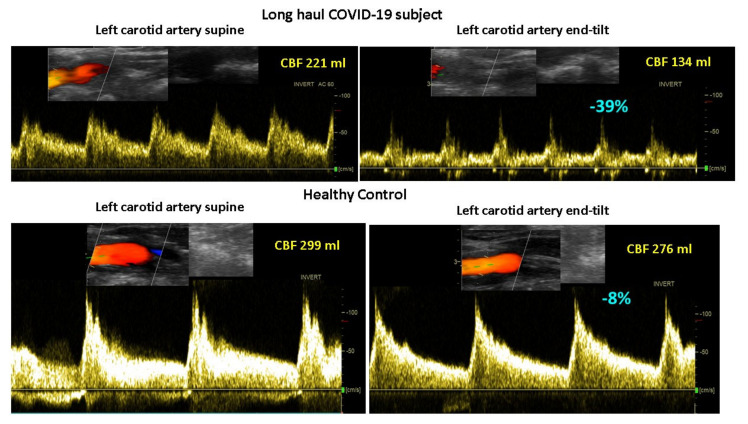
Example of cerebral blood flow images of the left carotid artery supine (**left** side) and end-tilt standing (**right** side) of a long-haul COVID-19 subject (**upper** panel) and a healthy control (**lower** panel). Legend Figure 4: CBF: cerebral blood flow.

**Figure 5 medicina-58-00028-f005:**
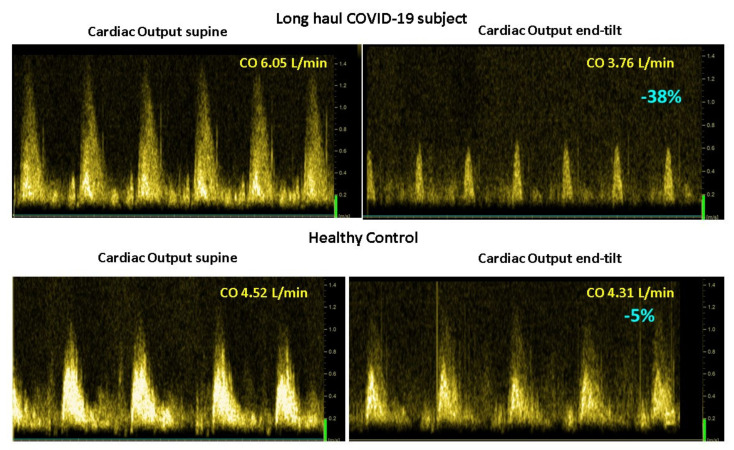
Example of cardiac output images supine (**left** side) and end-tilt standing (**right** side) of a long-haul COVID-19 subject (**upper** panel) and a healthy control (**lower** panel). Legend Figure 5: CO: cardiac output.

**Table 1 medicina-58-00028-t001:** Baseline characteristics.

	Healthy Controls (*n* = 20) Group 1	COVID-19 POTS (*n* = 10) Group 2	ME/CFS POTS (*n* = 20) Group 3	ME/CFS normHRBP (*n* = 20) Group 4	ANOVA and Post-Hoc Tukey
Male/female (*n*)	6/14	3/7	6/14	6/14	ns
Age (years)	30 (7)	30 (7)	30 (7)	30 (7)	F (3, 66) = 0.010; *p* = 0.99
Fulfilling typical ME criteria		9 (90%)	20 (100%)	18 (90%)	Chi-square 2128.2 *p* = 0.35
Fulfilling atypical ME criteria		1 (10%)	0 (0%)	2 (10%)	
Fulfilling CFS criteria		10 (100%)	20 (100%)	20 (100%)	ns
Fulfilling IOM criteria		10 (100%)	20 (100%)	20 (100%)	ns
Disease duration, years, #(range)	NA	1 (1–1.8)	9.5 (4–14.5)	10 (7–13.8)	X^2^(2) = 21.03; *p* < 0.0001. Post-hoc tests: 2 vs. 3 *p* = 0.0002; 2 vs. 4 *p* < 0.0001
BSA (m^2^)	1.88 (0.19)	1.85 (0.17)	1.90 (0.22)	1.75 (0.12)	F (3, 66) = 2.86; *p* = 0.048
BMI (kg/m^2^)	24.8 (4.5)	23.2 (5.4)	22.8 (3.8)	23.4 (4.1)	F (3, 66) = 0.79; *p* = 0.50

BMI: body mass index; BSA: body surface area; CFS: chronic fatigue syndrome; IOM: Institute of Medicine: ME: myalgic encephalomyelitis; normHRBP: normal heart rate and blood pressure response; POTS: postural orthostatic tachycardia syndrome; SEID: systemic exertion intolerance disease; # median (IQR) and Kruskal–Wallis test with Dunn’s multiple comparisons test BMI: body mass index; BSA: body surface area; CFS: chronic fatigue syndrome; ME: myalgic encephalomyelitis; normHRBP: normal heart rate and blood pressure response; POTS: postural orthostatic tachycardia syndrome; SEID: systemic exertion intolerance disease.

**Table 2 medicina-58-00028-t002:** Symptom cluster data of ME, CFS, and IOM ME/CFS criteria.

Symptom/Symptom Cluster	COVID-19 Cases % Present	95% CI COVID-19	ME/CFS Controls % Present	95% CI ME/CFS	*p*-Value #
Fatigue	100	100–100	100	100–100	1.0 *
Exercise intolerance physical/mental	100	100–100	97.5	93–102	1.0 *
PEM post physical/mental exercise	100	100–100	100	100–100	1.0 *
Prolonged recovery	100	100–100	100	100–100	1.0 *
Memory problems	90	71–109	92.5	84–101	0.80 #
Headache	100	100–100	92.5	84–101	1.0 *
Muscle pain	100	100–100	75	62–88	0.18 *
Joint pain	80	55–105	70	56–84	0.53 #
Unrefreshing sleep	100	100–100	100	100–100	1.0 *
Sensory hypersensitivity	100	100–100	87.5	77–98	0.57 *
Neuromotor abnormalities	100	100–100	92.5	84–101	1.0 *
Sore Throat	80	55–105	77.5	65–90	0.86 #
Tender lymph nodes	90	71–109	60	45–75	0.07 #
Viral infection hypersensitivity	70	42–98	57.5	42–73	0.47 #
Gastro-intestinal problems	100	100–100	90	81–99	0.57 *
Genito-urinary problems	60	30–90	42.5	27–58	0.32 #
Food/chemical hypersensitivity	40	10–70	75	62–88	0.03 #
Orthostatic intolerance	100	100–100	97.5	93–102	1.0 *
Respiratory symptoms	100	100–100	87.5	77–98	0.57 *
Thermal instability	80	55–105	72.5	59–86	0.63 #
Extreme temperature intolerance	90	71–109	77.5	65–90	0.50 #

95% CI: 95 percent confidence intervals; # Chi-square (2 × 2 table) or * Fisher’s exact test (if one of the columns shows 100%).

**Table 3 medicina-58-00028-t003:** Tilt table test data.

	Healthy Controls (*n* = 20) Group 1	COVID-19 POTS (*n* = 10) Group 2	ME/CFS POTS (*n* = 20) Group 3	ME/CFS normHRBP (*n* = 20) Group 4	ANOVA and Post-Hoc Tukeys
Heart rate supine (bpm)	68 (12)	73 (15)	78 (16)	79 (14)	F (3, 66) = 2.59; *p* = 0.06
Heart rate end-tilt (bpm)	79 (12)	108 (16)	113 (19)	92 (12)	F (3, 66) = 20.67; *p* < 0.0001. Post-hoc tests: 1 vs. 2 *p* < 0.0001; 1 vs. 3 *p* < 0.0001 and 3 vs. 4 *p* = 0.0001
Systolic BP supine (mmHg)	138 (18)	133 (16)	132 (24)	133 (17)	F (3, 66) = 0.42; *p* = 0.74
Systolic BP end-tilt (mmHg)	134 (15)	136 (15)	129 (18)	133 (17)	F (3, 66) = 0.64; *p* = 0.59
Diastolic BP supine (mmHg)	79 (10)	85 (12)	81 (19)	78 (12)	F (3, 66) = 0.68; *p* = 0.57
Diastolic BP end-tilt (mmHg)	83 (8)	100 (11)	89 (17)	86 (8)	F (3, 66) = 4.68; *p* = 0.0051. Post-hoc tests: 1 vs. 2 *p* = 0.0032
CBF supine (mL/min)	617 (83)	629 (70)	637 (121)	605 (86)	F (3, 66) = 0.43; *p* = 0.73
CBF end-tilt (mL/min)	591 (84)	418 (64)	455 (95)	451 (69)	F (3, 66) = 15.74; *p* < 0.0001. Post-hoc tests: 1 vs. 2 *p* < 0.0001; 1 vs. 3 *p* < 0.0001 and 1 vs. 4 *p* < 0.0001
P_ET_CO_2_ supine (mmHg)	37 (2)	39 (3)	38 (3)	37 (2)	F (3, 66) = 1.76; *p* = 0.16
P_ET_CO_2_ end-tilt (mmHg)	36 (2)	29 (3)	27 (6)	29 (4)	F (3, 66) = 16.19; *p* = 0.0002. Post-hoc tests: 1 vs. 2 *p* < 0.0001; 1 vs. 3 *p* < 0.0001 and 1 vs. 4 *p* < 0.0001
Delta P_ET_CO_2_ (mmHg)	−1 (1)	−10 (3)	−10 (3)	−8 (4)	F (3, 66) = 37.18; *p* < 0.0001. Post-hoc tests: 1 vs. 2 *p* < 0.0001; 1 vs. 3 *p* < 0.0001 and 1 vs. 4 *p* < 0.0001
CI supine (L/min/m^2^)	2.29 (0.30)	2.86 (0.36)	2.82 (0.46)	2.65 (0.39)	F (3, 66) = 8.08; *p* = 0.0001. Post-hoc tests: 1 vs. 2 *p* = 0.0016; 1 vs. 3 *p* = 0.0003
CI end-tilt (L/min/m^2^)	2.08 (0.24)	2.20 (0.54)	2.19 (0.48)	1.99 (0.30)	F (3, 66) = 1.25; *p* = 0.30
Perc reduction CI (%)	−9 (5)	−23 (14)	−22 (12)	−25 (5)	F (3, 66) = 12.22; *p* < 0.0001. Post-hoc tests: 1 vs. 2 *p* = 0.0008; 1 vs. 3 *p* = 0.0002 and 1 vs. 4 *p* < 0.0001

BP: blood pressure; CBF: cerebral blood flow; CI: cardiac index (cardiac output indexed for body surface area); normHRBP: normal heart rate and blood pressure response; POTS: postural orthostatic tachycardia syndrome; P_et_CO_2_: end tidal carbondioxide pressure.

## Data Availability

The data presented in this study are available on request from the corresponding author. The data are not publicly available due to privacy reasons.

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
