# Peer review of "Orthostatic Symptoms and Reductions in Cerebral Blood Flow in Long-Haul COVID-19 Patients: Similarities with Myalgic Encephalomyelitis/Chronic Fatigue Syndrome"

_medicina, 2021, doi:10.3390/medicina58010028_

Round 1

Reviewer 1 Report

GENERAL COMMENTS:

The submitted manuscript is describing findings in a small cohort of long COVID patients and comparing them to patients with chronic fatigue syndrome and healthy controls. The manuscript would only need few minor improvements: 

  1.      The main limitation of the study is a small number of included subjects, particularly long COVID patients.
  2.      The study is otherwise very interesting and it is also very relevant.
  3.       It was well designed and performed.
  4.       The study is also well written.  

SPECIFIC COMMENTS

  1. Title: Somehow long –would be nice to shorten it.
  2.  Abstract: Long, but fine.
  3. Introduction: Well written.
  4. Methods: Extensive, but well described.
  5. Results: Style of this section should be changed in order to shorten it. Paragraphs start with “Figure X shows” instead of straight description of findings, followed by reference to Figures. Provide just findings, without explanations in this section.
  6. Discussion: Fine.
  7. Table 1: Height was mentioned, but I miss values in the table.
  8.  Legends to table 1 include duplication of the text.
  9. Figures are very informative.

Author Response

See responses in report.

Reviewer 2 Report

The manuscript presented for the review is very interesting and shed a light on the possible complications of COVID19 disease. The Authors indicated similarities in symptoms of long-haul SARS-CoV-2 viral infection and myalgic encephalomyelitis/chronic fatigue syndrome (ME/CFS). They suggest that SARS-CoV-2 may act as a trigger of ME/CFS. Despite the limitations mentioned by the Authors, the experience was prepared carefully and the aim of the study is clear and comprehensible. The introduction briefly highlight the study assumptions. Materials and methods are described in a great detail. The Authors provide an interesting and accurate description of the results.

As the reviewer, I have several comments about the manuscript:

  • The number of words in the abstract exceeds the maximum number allowed, which is mentioned in “Instructions for authors”.
  • According to “Instructions for authors” in the abstract, a Background and Objectives should be placed instead of Introduction.
  • Ordinal numbers in keywords are not necessary.
  • At the end of abstract (line 33), there are two dots.
  • In line 92, there is an IOM abbreviation which appear for the first time and is not explained. The Authors should decide if they use a full name or abbreviation in the text (lines: 92, 103, 109, 111, table 1, 220, 240, 243, 325, 342).
  • In lines 155, 255, 260, 265 in the CO2 the “2” digit should be in subscript as in table 3.
  • In line 192, abbreviation SD should be explained. Mean with standard deviation (SD). Same in line 193. There is no explanation of IQR abbreviation. Median with interquartile range (IQR).
  • The statistically significant differences in figure 2 and 3 between Healthy controls and the other experimental groups should be marked on the graph.
  • In lines 390, 393… the VO2 abbreviation should be explained. Is that a volume of oxygen uptake?
  • The references should be corrected according to “Instructions of authors”. All references do not fulfil the requirements.
  • Reference position 71 is not present in the text.

In my opinion, the manuscript after minor revision will be suitable for publication in Medicina.

Author Response

See comments in report.
